# Comparison of traditional methods versus SAFEcount for filling prescriptions: A pilot study of an innovative pill counting solution in eSwatini

Paul J. Krezanoski[1,2]*, Joseph D. Krezanoski[2,3], Nkosinathi Nxumalo[4], Rose Gabert[2,5], Alison B. Comfort[1,2], Phinda Khumalo[6], Kidwell Matshotyana[7]

1 University of California San Francisco, San Francisco, CA, United States of America, 2 Opportunity Solutions International, San Francisco, CA, United States of America, 3 Massachusetts General Hospital Institute of Health Professions, Boston, MA, United States of America, 4 ICAP at Columbia University, Mbabane, eSwatini, 5 University of Washington, Seattle, WA, United States of America, 6 National Yang Ming University, Taipei, Taiwan, 7 Eastern Cape Department of Health, Bhisho, South Africa

* paul.krezanoski@ucsf.edu

**Data Availability Statement:** The datasets used and/or analyzed during the current study are

## Abstract

### Background

Packaging medications is a crucial component of health system efficiency and quality. In developing countries, medications often arrive in bulk containers that need to be counted by hand. Traditional counting is time-consuming, inaccurate and tedious. SAFEcount is a novel and inexpensive handheld device that may improve the accuracy and speed of pill-counting in resource limited settings. We designed a head-to-head trial to compare traditional and SAFEcount prescription filling in eSwatini.

### Methods

We recruited 31 participants from 13 health facilities throughout eSwatini. Speed and accuracy for each prescription was recorded while each participant filled prescriptions of various quantities using both the traditional and SAFEcount methods.

### Results

Traditional pill counting resulted in an error rate of 12.6% inaccurate prescriptions compared to 4.8% for SAFEcount (p<0.0001). SAFEcount was 42.3% faster than traditional counting (99.9 pills per minute versus 70.2; p<0.0001). Using SAFEcount was preferred over traditional pill counting by 97% (29/30) of participants.

### Conclusions

The SAFEcount device is a preferred alternative by counting personnel and is significantly faster and more accurate compared to traditional counting methods. SAFEcount could help improve the efficiency and quality of health care delivery in place of traditional hand counting.

available at the Open Science Framework (OSF) public data repository via: https://osf.io/38nyj/.

**Funding:** This study was funded by Opportunity Solutions International Incorporated, a 501(c)3 non-profit organization in San Francisco, California, USA.

**Competing interests:** PJK and ABC are Founding Directors and members of the Board of Opportunity Solutions International (unpaid). JDK is an unpaid employee of Opportunity Solutions International and the inventor of SAFEcount. Opportunity Solutions International and JDK are co-owners of intellectual property related to a pending patent for the SAFEcount device (US Patent Application No. 16/208,525). This does not alter our adherence to PLOS ONE policies on sharing data and materials.

## Introduction

Improving the quality and efficiency of health care in low-resource countries has been identified as a crucial step towards achieving the World Health Organization's Sustainable Development Goals by 2030 [1]. The packaging and provision of medications to patients at health clinics, via the pharmaceutical dispensary, is increasingly being recognized as a crucial component of overall health system efficiency and quality [2,3]. In eSwatini, there are only approximately 240 public and private health facilities serving a population of 1.3 million people, making patient volume at health facilities a significant challenge [4]. In particular, a large proportion of patients are HIV positive, 27.4% by 2017 estimates [5]. Many of these patients require daily co-trimoxazole (CTX) as prophylaxis against opportunistic infections until clinically stable on anti-retroviral treatment [6,7].

In eSwatini, as in the majority of dispensaries throughout the world, pills like CTX typically arrive from the Central Medical Store (CMS) in bulk containers (e.g. 1000 pills) and then require individual counting and placing into packaging for dispensing to patients. This individual counting of prescriptions typically occurs by hand using a tray and spatula. We call this method the traditional counting method. Counting by hand is prevalent throughout less developed countries because the infrastructure and resources are inadequate to provide for the typically electronic pill counting solutions used commonly in middle and higher income countries [8]. In eSwatini, for example, there were only a few electronic pill counting machines in operation in the entire country when this study was performed. These machines were only at large hospitals and they often had reliability or servicing issues (personal conversations). In addition, compared to bulk containers, patient-ready blister packs of medications are more expensive, do not allow for flexible prescription quantities and the extra materials used in the packaging increases the physical space necessary for delivery and storage.

Traditional prescription counting represents an important bottleneck in the medication dispensary process because it is time-consuming, inaccurate and tedious. CTX prescriptions, for example, are typically provided in quantities of 30, 60 or 120. Hand counting these large quantity prescriptions takes significant personnel hours. Staff engaged in counting and packaging pills may be more profitably utilized in direct patient care or managing stocks of supplies. Alternatively, if the extra hours spend doing traditional counting could be avoided, then the surplus personnel costs could be averted altogether by hiring less clinic staff. Some facilities tend to use cleaners and other auxiliary health staff to count and pre-package the medicines. Furthermore, excessive time spent counting and pre-packaging pills may lead to increased patient wait times, decreased patient satisfaction and increased staff burnout related to job obligations that require tedious manual tasks [8,9].

In addition to being time-consuming, pill counting by hand has a high rate of counting errors. When more pills are provided in the pill bottle to patients than are prescribed ("overcount"), there are problems with wasted pills and potential toxicity for patients. When there are fewer pills in the prescription than prescribed ("undercount"), patients are exposed to negative health outcomes from under treatment, such as increased risk of opportunistic infections in the case of under treatment with CTX.

Finally, for both overcounting and undercounting of prescriptions, there are implications for tracking and ensuring patient adherence and retention. Most facilities use pill counts when patients return for refills of their medications as a measure of adherence. If the prescriptions are miscounted to begin with, this creates an unnecessarily adversarial environment between patient and provider with negative implications for patient outcomes [10]. If a large proportion of prescriptions are overcounted or undercounted, this may call into question the value of pill counts as an accurate means of assessing patient adherence to prescribed drug regimens.

In 2012 we invented SAFEcount, a robust, easy-to-use, and inexpensive all-plastic handheld device for counting pills quickly and accurately (United States Provisional Patent Application Serial No. 62/593,869, filed December 1, 2017). SAFEcount was designed to be far less expensive than electronic pill counters and far more accurate and rapid than traditional pill counting. SAFEcount is designed as a "mid-level" medical device that represents a technological upgrade for dispensaries specifically in low- to middle-income settings. SAFEcount uses interchangeable grids, with enough wells for the required prescription count and wells shaped in the size of the pill of interest (S1, S2 and S3 Photographs). When pills are poured over the top, a swirling motion of the user's hand captures the exact number of pills necessary to fill a prescription in the grid and shunts away excess pills for future use. Using a trap-door underneath the grid, activated by pushing the system down on a tabletop, the pills fall into an adaptable handle which acts as a funnel for the seamless transfer of the pills to bags for provision to the patient. (For video demonstrating functionality: https://www.youtube.com/watch?v=lGhi9U7zgfc) The grids can be made to function for all tablet shapes, sizes and dosages. SAFEcount was specifically designed to balance simplicity and ease-of-use. To date, we have built four prototypes, most recently using 3D printing technology.

This study was conceived as a head-to-head comparison of traditional versus SAFEcount counting among regular pill counters in eSwatini. We used CTX as our paradigm example of recounted bulk pills, with its typical 30, 60 and 120 pill prescriptions. We designed an experiment, separate from clinic activities (using dummy CTX pills not for patient consumption), and provided prizes to incentivize both speed and accuracy. The primary goal of the study was to compare both the speed and accuracy of the traditional and SAFEcount pill counting methods and provide pilot data on the potential of SAFEcount to act as a novel and affordable pill counting tool to improve health care in eSwatini and beyond.

## Methods

The study was performed in July-August 2016 in eSwatini. The study protocol was approved by eSwatini's Scientific and Ethics Committee at the Ministry of Health (IRB # 000–9688) and by Partners Human Research Committee at Massachusetts General Hospital (IRB # 2016P001099/MGH). Fifteen health facilities were sampled from a list of health facilities provided by the eSwatini Ministry of Health. We narrowed the list to 124 public or private health facilities that served the general population (i.e. non-specialty [e.g. TB, HIV] or special population [e.g. police, university, prison] facilities). Facilities were included if they served at least 500 patients per month. First, we selected 11 facilities from each of three groups defined by monthly patient volumes: small (500–1100 patients per month), medium (1101–2200), and large (>2200). Of these 11 facilities, we then chose 5 at random from each stratum, while balancing representation from the four regions in eSwatini. The final list of 15 health facilities represented all four regions: six from Hhohho (two small, one medium and three large), five from Manzini (three small and two medium), three from Lombombo (one medium and two large) and one from Shiselweni (medium). However, due to time constraints, only 13 facilities were visited (one less from both Hhohho [small] and Manzini [small]).

On arrival at each health facility, we secured a separate location in which to perform the experiment, usually an empty room with a table. The materials for the protocol included bulk pill bottles of our own CTX tablets (both round-shaped 480 mg and oblong-shaped 960 mg), trays and spatulas for traditional counting and SAFEcount devices with the appropriate grids. We then described the study to the facility supervisor, or other in-charge present at the time of our visit, and requested a list of staff available that day who count pills as part of their job responsibilities. Individual pill counters were enrolled if they counted pills for patient

prescriptions at least twice in the last month. Exclusion criteria included individuals under 18 years of age and those who reported an allergy to CTX when prompted during the consent process. Each of the participants was brought to the separate room when it was convenient for them depending on their daily duties and written consent was obtained before study commencement.

Data was collected for each participant on the age, gender, education level, job title and their pill counting experience. We collected data on participant attitudes about pill counting, perceptions of the clinic burden of pill counting and acceptability of the SAFEcount device following completion of the counting protocol. The amount of time it took for batches of prescriptions to be counted and the accuracy of individual prescriptions were recorded in real-time by two research assistants during the pill counting activities. We also video recorded and took pictures of every pill counting activity for later validation of the manual records of speed and accuracy and to review participant handling of the device.

The primary outcome for the study was to compare traditional hand counting with a tray and spatula to SAFEcount according to: 1) the accuracy of the prescriptions as measured by the proportion of prescriptions with errors compared to the expected pill count and 2) the counting speed measured as the average pills counted per minute for each participant. Secondary outcomes included other relevant clinic-wide or supply chain metrics, such as average magnitude of errors and average number of pills wasted, and the acceptability of counting pills with traditional method versus SAFEcount. The preceding pill counting accuracy metrics are focused on accuracy at the level of a patient prescription. Another way to assess the accuracy of pill counting is to calculate a rate of error per 1000 pills counted (per standard bottle of CTX). This metric captures the total amount that a pharmaceutical dispensary would either waste (more pill provided than prescribed, i.e. overcounting) or accidentally retain (fewer pills provided than prescribed, i.e. undercounting) as a result of pill counting errors.

The pill counting protocol was designed for a controlled setting. The controlled setting was deemed necessary so as to limit interference with the regular functioning of the health facility and not to contaminate medications that were to be consumed by patients (as mentioned, we provided our own tablets for the counting protocol). None of the study participants had any prior experience with the SAFEcount device. The counting protocol involved a short practice session to become familiar with SAFEcount (12 prescriptions total) followed by the speed- and accuracy-measured activities alternating between traditional and SAFEcount counting (Table 1). During the counting procedure, one research assistant recorded the prescription

**Table 1. Protocol for counting activities.**

| Step | Activity | Prescriptions, number and type of tablets |
|---|---|---|
| 1 | Practice SAFEcount | 5 prescriptions of 30 oblong tablets |
| 2 | Practice SAFEcount | 5 prescriptions of 60 round tablets |
| 3 | Practice SAFEcount | 2 prescriptions of 120 round tablets |
| 30 pill prescriptions | | |
| 4 | Traditional Counting | 15 prescriptions of 30 oblong tablets |
| 5 | SAFEcount Counting | 15 prescriptions of 30 oblong tablets |
| 60 pill prescriptions | | |
| 6 | Traditional Counting | 15 prescriptions of 60 round tablets |
| 7 | SAFEcount Counting | 15 prescriptions of 60 round tablets |
| 120 pill prescriptions | | |
| 8 | Traditional Counting | 5 prescriptions of 120 round tablets |
| 9 | SAFEcount Counting | 5 prescriptions of 120 round tablets |

speeds, from pouring pills to sealing the prescription bag. Counting accuracy was determined by a second research assistant using a specially designed SAFEcount grid that was walled off to capture all pills from the pill bag. A second verification count was performed whenever an error was detected.

We imposed a few conditions in order to make the study as realistic as possible despite the controlled setting. First, the participants were required to stand at a table, as this seemed to be the norm in the dispensaries we had visited where seating was minimal. Second, following the practice session with the SAFEcount device at the beginning of the protocol, the rest of the counting activities proceeded uninterrupted. The complete protocol of uninterrupted pill counting thus lasted about 2 hours, depending on the participant speed, and comprised 3,900 pills counted. These rules were implemented in order to replicate as much as possible the mental and physical fatigue that arises when pill counting occurs in the dispensary setting.

In the presence of these mental and physical stresses, real-life pill counting must also balance two fundamental constraints. Pill counters want to count as quickly as possible (speed), but this needs to be moderated by making the counting correct (accuracy). The ideal pill counter is as fast and as accurate as possible. In real life, given that medication errors can harm patients, accuracy should be as close to 100% as is feasible. Thus, our third imposed condition to simulate dispensary conditions in our controlled protocol was to incentivize both speed and accuracy equally. To achieve this, we showed the participants an incentive payout table (Fig 1) which divided speed and accuracy into four quadrants based on median performance. Since we did not yet have calculated values for median performance, at the completion of the counting activities we gave every participant prepaid cellphone airtime of 60 Swazi Lilangeni (SZL; approximately $4.28 in August 2016) (yellow boxes in Fig 1).

All data analyses were performed using Stata 14 (StataCorp. 2016. Release 14. College Station, TX). Summary statistics are reported for participant demographic characteristics, pill counting experience and Likert scales of attitudes about and perceptions of pill counting. The speed and accuracy of traditional and SAFEcount pill counting are reported with means and proportions compared using two-sided t-tests and differences reported with 95% confidence intervals (CIs).

## Results

Thirty-one participants were enrolled from the 13 clinics visited. Seventy-one percent (22/31) were women with an average age of 39 years. Most attended some high school and 19% had completed university. Four nurses (13%), 12 HIV counselors (39%) and two pharmacy personnel were included, with the remaining thirteen reporting their role as a clinic "cleaner" (33%) or other support staff (10%). Most participants (19/31) estimated that they spend more than 40% of their effort counting pills, with five (16%) reporting that they spend 80–100% effort on pill counting. Most reported that they counted up to 90,000 pills of CTX per week (81%), but three participants (11%) estimated that they counted between 121,000–150,000 pills CTX per week (Table 2). Overall, participants reported spending an average of 3.58 hours per day counting pills.

Most participants found counting pills "exhausting" or strongly disagreed with the statement that pill counting "does not make them tired". Perceptions were split on questions about whether pill counting interferes with clinic activities or whether time spent pill counting could be used for patient care (Table 3).

A total of 2,170 prescriptions were counted using both the traditional method and SAFEcount. The proportion of prescriptions with errors by traditional counting was 12.6% (137/1085) compared to 4.8% (52/1085) with SAFEcount (p<0.0001) (Table 4). Prescriptions are

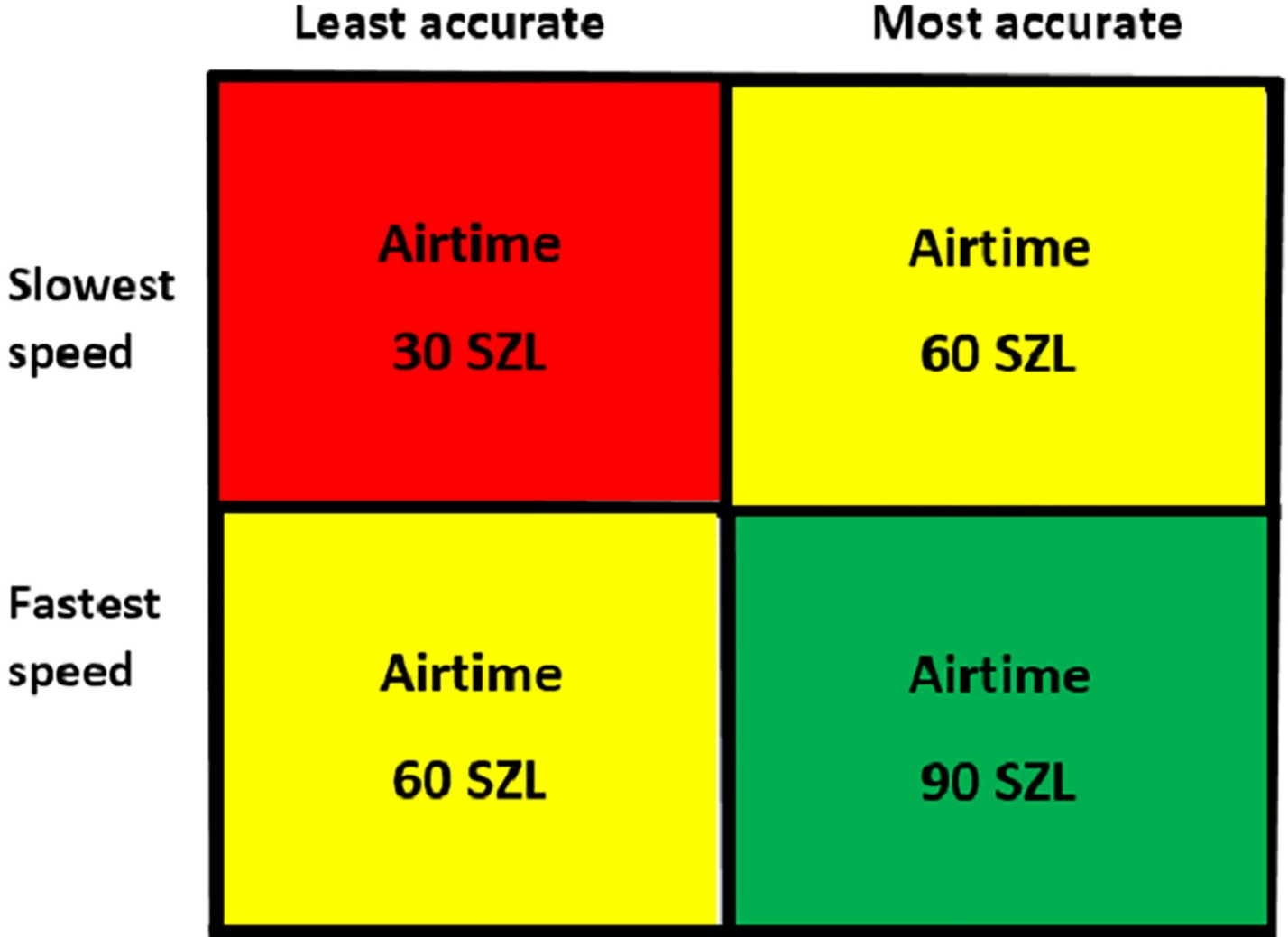

**Fig 1. Payout table for SAFEcount pilot.**

2.6 times less likely to be counted in error using SAFEcount compared to traditional counting. Next we explored whether the size of the prescription being counted impacted the likelihood of an error. There was no significant difference in prevalence of errors between counting with the traditional method and SAFEcount for 30 pill prescriptions, but for 60 and 120 pill prescriptions SAFEcount counting resulted in significantly lower prescription errors (Fig 2).

When errors occurred, the average magnitude of pills counted in error (absolute value of additional pills provided in error [overcounted] or pills retained in error [undercounted]) was significantly higher with SAFEcount: 6.7 pills versus 2.8 pills via traditional counting, a difference of 3.9 pills (p = 0.008) (Table 4). Sixty-two percent of the traditional method errors were overcounted prescriptions versus 42% for SAFEcount (p = 0.014). We also explored the distribution of the magnitude of errors by the type of counting method used and size of the prescription. For the traditional counting errors, all incorrect prescriptions appear within +/- 20 pills. For SAFEcount errors, a much smaller frequency of prescriptions is seen at fewer than +/-5 pills and several other clusters appear at +/- 15 and +/- 60 pills. These larger clusters only occurred when counting out 120 pill prescriptions (Fig 3).

**Table 2. Participant characteristics.**

| Participant sex | | | |
|---|---|---|---|
| | Female | 22 | 71.0% |
| | Male | 9 | 29.0% |
| Participant age | | | |
| | ≤ 35 | 11 | 35.5% |
| | 36–45 | 14 | 45.2% |
| | 46 + | 6 | 19.4% |
| Highest level of education completed | | | |
| | None | 2 | 6.5% |
| | Primary | 2 | 6.5% |
| | Some high school | 12 | 38.7% |
| | Completed high school | 9 | 29.0% |
| | Some university | 1 | 3.2% |
| | Completed university | 6 | 19.4% |
| Participant position title | | | |
| | Nurse | 4 | 12.9% |
| | HIV Counselor | 12 | 38.7% |
| | Cleaner | 10 | 32.3% |
| | Pharmacy personnel | 2 | 6.5% |
| | Other | 3 | 9.7% |
| Estimated % effort spent counting pills | | | |
| | 0–20% | 7 | 22.6% |
| | 20–40% | 5 | 16.1% |
| | 40–60% | 10 | 32.3% |
| | 60–80% | 4 | 12.9% |
| | 80–100% | 5 | 16.1% |
| Average bottles of CTX bagged/week per participant | | | |
| | 0–30 | 17 | 61% |
| | 31–60 | 4 | 14% |
| | 61–90 | 4 | 14% |
| | 91–120 | 0 | 0% |
| | 121–150 | 3 | 11% |
| | 151+ | 0 | 0% |

A total of 60,450 pills were counted in the study, 30,225 pills each via the traditional method and SAFEcount. Per 1000 pills (or per bottle of CTX), there were 3.3 wasted pills per 1000 using traditional counting and 0.12 improperly retained pills per 1000 with SAFEcount.

Next we examined the effect of SAFEcount on the speed of counting. Overall, counting via SAFEcount resulted in an average participant counting rate of 99.9 pills per minute compared to 70.2 pills per minute for traditional counting. This difference of 29.7 pills per minute represents a 42% increase in pill counting speed using SAFEcount compared to the current traditional counting method ($p < 0.0001$). (Table 5)

Given that the ideal pill counter maximizes both accuracy of prescriptions and speed of counting, we have plotted in Fig 4 the speed (pills per minute) and accuracy (proportion of prescription errors) for each of the thirty-one participants by the type of counting method. It is clear from this plot that SAFEcount leads to significant improvements in both speed and accuracy, clustering individuals towards the bottom right of the plot representing faster counting speeds and fewer errors per prescription.

**Table 3. Perceptions about pill counting.**

|  | Strongly disagree | Disagree | Neutral | Agree | Strongly Agree | Total |
|---|---|---|---|---|---|---|
| I find counting pills exhausting | 2 | 0 | 2 | 2 | 24 | 30 |
| I have enough time to count all the pills needed for patients during a typical day | 12 | 0 | 1 | 4 | 13 | 30 |
| Counting pills does not make me tired | 21 | 0 | 1 | 1 | 6 | 29 |
| I want a faster way to prepackage prescriptions | 0 | 0 | 0 | 0 | 29 | 29 |
| I try to count prepacked prescriptions as fast as possible | 0 | 0 | 4 | 5 | 21 | 30 |
| My time should be spent on tasks other that than pill counting | 11 | 1 | 2 | 2 | 14 | 30 |
| Our clinic spends many extra hours counting pills that could be used for helping patients | 14 | 0 | 3 | 2 | 11 | 30 |

Note: Totals reflect those that responded to the particular question

Data on the acceptability of SAFEcount are presented in Table 6. On verbal responses, all participants reported that they would choose SAFEcount over traditional pill counting, all agree that SAFEcount made counting easier and all report that they could see themselves using SAFEcount to count pills.

## Discussion

This head-to-head comparison study was designed to test the speed and accuracy of pill counting using a traditional hand counting method versus SAFEcount, a novel handheld pill counting device. The results show that SAFEcount was 2.6 times less likely to lead to a prescription error and increased pill counting speed by 29.7 pills per minute, representing a 42% improvement. These results demonstrate that SAFEcount could be a valuable tool for improving the quality of pharmaceutical dispensing in resource constrained settings where traditional hand counting of bulk prescriptions remains the norm.

Our results suggest that SAFEcount could have a large impact on clinic quality of patient care, patient safety and labor costs. For example, reducing prescription error rates from 12.6% with traditional counting to 4.8% with SAFEcount would result in 8 more patients out of every 100 receiving correct prescriptions. The 3.3 pills overcounted per 1000 when using traditional counting represents a wastage of 3 pills per bottle of CTX. Among our participants, 36% reported counting at least 31,000 or more pills of CTX per week. Applying the error rate above, due to traditional counting, these participants would waste at least 102 pills (3.3 wasted/bottle * 31 bottles) per week, or nearly 5.3 bottles per pill counter per year (102 pills/week * 52 weeks/year).

**Table 4. Accuracy of pill counting using traditional method versus SAFEcount.**

|  | N | Traditional method | SAFECount | Difference | 95% two-sided p-value |
|---|---|---|---|---|---|
| Proportion of prescriptions with error (95% CI) | 2,170 | 12.6% (10.6% - 14.6%) | 4.8% (3.5% - 6.1%) | -7.80% (-10.2% —-5.5%) | <0.0001 |
| Total prescription errors (%) | 189 | 137 (72.5%) | 52 (27.5%) |  |  |
| Average magnitude (absolute value) of prescription error (95% CI) | 189 | 2.8 (2.3–3.3) | 6.7 (2.1–11.2) | 3.9 (1.0–6.7) | 0.0079 |
| Prescription errors **over** counted (n[%]) | 107 | 85 (62.0%) | 22 (42.3%) | -19.7% (-35.5% —-4.0%) | 0.0144 |
| Average magnitude of **over** count errors (95% CI) | 107 | 2.8 (2.2–3.5) | 7.7 (-0.2–15.6) | 4.9 (0.9–8.9) | 0.0169 |
| Prescription errors **under** counted (n[%]) | 82 | 52 (38.0%) | 30 (57.7%) | 19.7% (35.5% - 4.0%) | 0.0144 |
| Average magnitude of **under** count errors (95% CI) | 82 | 2.7 (2.0–3.4) | 5.9 (0.2–11.6) | 3.2 (-1.1–7.5) | 0.1479 |

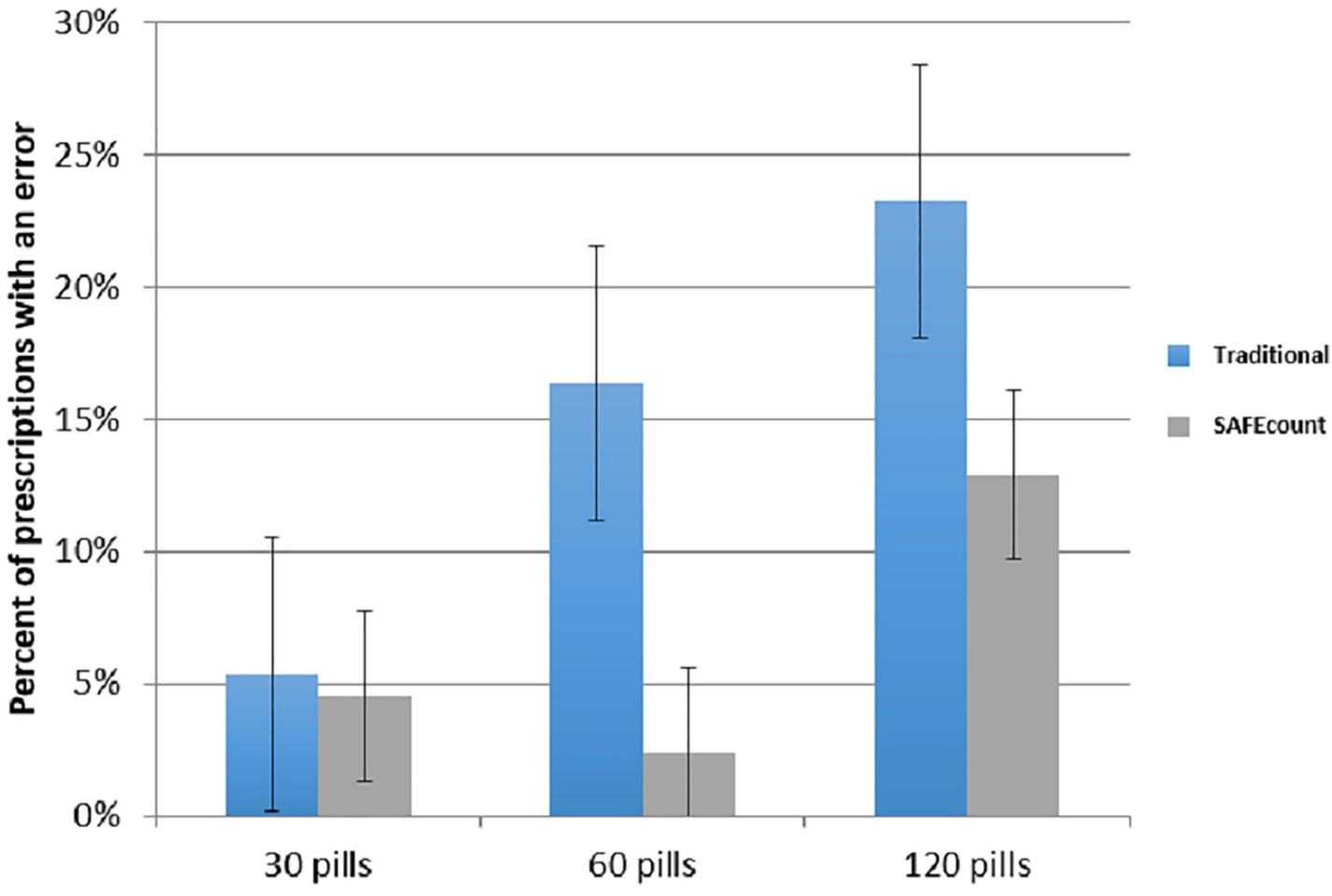

**Fig 2. Percent of prescriptions with an error by prescription volume and counting method.**

In addition, allowing pill counters to perform their roles 42% faster could lead to significant personnel cost savings. Overall the study participants reported spending an average of 3.58 hours per day on pill counting or nearly 18 hours per week. Based on current pill counting obligations, our estimates suggest that using SAFEcount could save more than 10 pill counting hours per pill counter per week, which represents 2.1 days worth of pill counting effort per week.

Finally, there is no reason to expect that our results are specific to CTX, which is a relatively less expensive medication. Since the grids are customizable for any pill shape or dosage and any prescription size, medicine dispensaries that count any medication in bulk would benefit. Relatively more expensive medications on essential medication lists, such as anti-hypertensive, anti-tuberculosis and diabetic medications represent leading candidates for SAFEcount related cost savings [11].

Our surveys of pill counter perceptions of pill counting activities demonstrate that most find the task tiring. There was an interesting split over whether participants considered pill counting as an area where efficiencies could allow for more time to care for patients or other clinic tasks. This finding appears to represent a split between participants whose main roles rely on pill counting. Indeed, participants whose main title is "Cleaner" but who perform pill counting as an additional task tend to believe that pill counting is essential and that there are

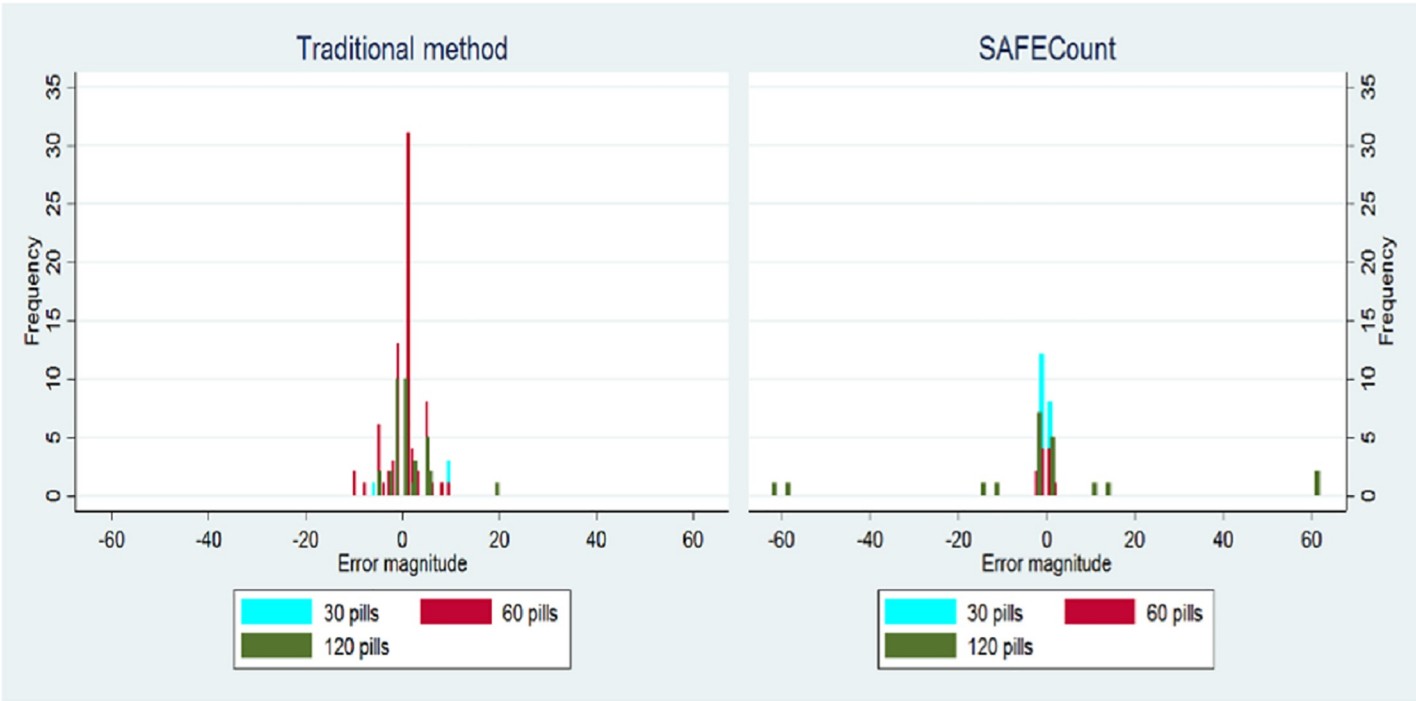

**Fig 3. Distribution of magnitude of prescription errors.**

no significant opportunity costs to counting. Participants with professional positions (nurse, pharmacy personnel) tend to identify pill counting as something that may be avoided, thus the time savings from SAFEcount may allow professional staff to focus on direct patient care while leaving pill counting tasks to non-professional positions.

In terms of the acceptability of SAFEcount in this experiment, SAFEcount was universally preferred over traditional methods of counting. Every participant reported that they enjoyed using SAFEcount and would prefer to continue to use it for counting activities in the future.

The benefits achieved by SAFEcount must be weighed against this study's findings that while there are fewer overall prescription errors with SAFEcount, the errors tend to be larger than hand counting: 6.7 pills versus 2.8 pills respectively (p<0.01). This greater average error magnitude is driven by rarer but larger errors that occurred with SAFEcount, with the distribution demonstrating clusters around 15 and 60 pill over or undercounts of 120 pill prescriptions. These are large errors and we believe they are unlikely to be a practical concern for two main reasons. First, in a clinic setting without incentivized time pressures of this experiment, large errors, such as an additional or missing 60 pills from a 120 pill prescription, would likely

**Table 5. Pill counting speed using traditional method versus SAFEcount.**

|  | N | Traditional method | SAFECount | Difference | 95% two-sided p-value |
|---|---|---|---|---|---|
| Average overall speed (pills per minute) (95% CI) | 31 | 70.2 (63.8–76.7) | 99.9 (92.9–107.0) | 29.7 (20.3–39.1) | <0.0001 |
| Average speed (pills per minute) by quantity: |  |  |  |  |  |
| 30 pill prescriptions | 31 | 58.1 (52.7–63.6) | 54.4 (50.5–58.3) | -3.7 (-10.3–2.9) | 0.2641 |
| 60 pill prescriptions | 31 | 74.7 (67.7–81.7) | 117.6 (109.1–126.1) | 42.9 (32.1–53.7) | <0.0001 |
| 120 pill prescriptions | 31 | 77.8 (70.1–85.6) | 127.7 (116.3–139.2) | 49.9 (36.4–63.4) | <0.0001 |

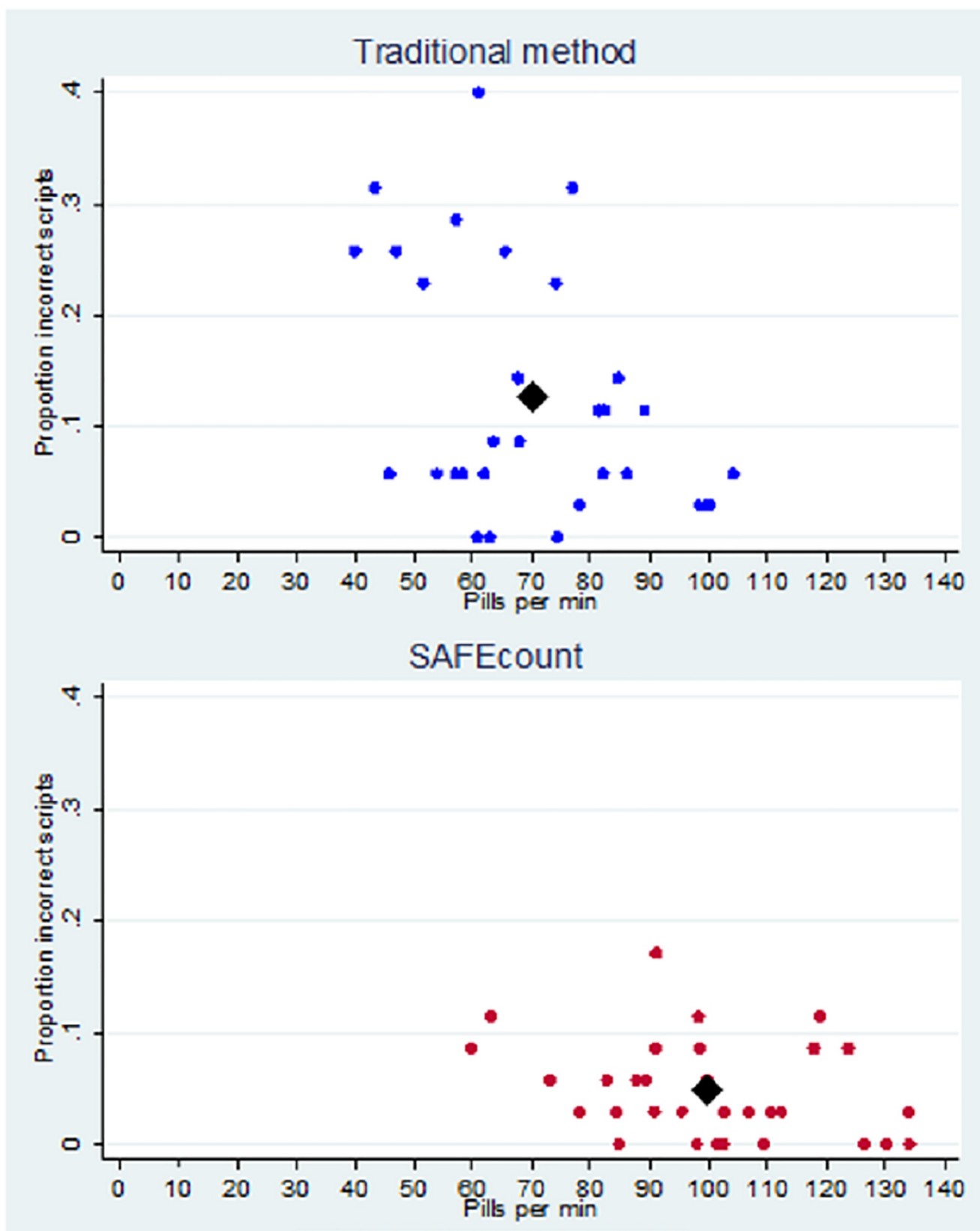

**Fig 4. Participant counting accuracy versus speed.**

**Table 6. SAFEcount acceptability.**

| | Strongly disagree | Disagree | Neutral | Agree | Strongly Agree | Total |
|---|---|---|---|---|---|---|
| I would choose SAFEcount to count tablets instead of using a plastic dish and spatula | 0 | 0 | 1 | 0 | 29 | 30 |
| Compared to manually counting pills, SAFEcount made counting tablets easier | 0 | 0 | 0 | 1 | 29 | 30 |
| I would rather manually count tablets with a plastic dish and spatula than use SAFEcount | 30 | 0 | 0 | 0 | 0 | 30 |
| I can see myself using the SAFEcount device while prepackaging prescriptions | 0 | 0 | 0 | 0 | 30 | 30 |

Note: Totals reflect those that responded to the particular question

be identified before pills are provided to patients. Second, the 120 pill prescription counting method for SAFEcount required two "pour and bag" cycles using a 60 pill grid. Our review of video recordings where errors were identified show that these errors typically were caused by an incomplete dumping of the pills from the handle after the first pour cycle, i.e. 15 pills remained in the device. This explains why a -15 pill deficit in one prescription of 120 pills was followed sequentially by a +15 pill surplus in the next 120 pill prescription. We have begun to explore ways to make device improvements to avoid these types of errors in the future.

One limitation of generalizing the results from this study is that it was performed in a controlled setting. A controlled setting was chosen in order to compare the efficacy of the two methods and control for external factors, such as interruptions, that may affect the accuracy and speed of pill counting in dispensaries. We do not believe there are significant reasons why the time and quality improvements identified in this study should not translate to the clinic setting. Another limitation of this study is that participants had limited exposure practice with SAFEcount (12 prescriptions of varying sizes; approximately 15 minutes) before counting and being recorded and this study was the first time that they were using the device. As such, this limitation would likely bias our estimates of the gains towards even faster and more accurate counting with SAFEcount as individuals become more familiar with the device. Hence, the large errors with SAFEcount may only be an indicator of inexperience, as learning rates likely vary between individuals. In scaling the device, we would expect that a longer training period, even one hour, for example, which could be self-directed, would go a long way towards decreasing errors and increasing speed even further. Our incentive scheme, whereby we attempted to provide an external motivation to participants to count as quickly and accurately as possible, may not match real-life incentives. This scheme was chosen because we wanted to avoid situations where individuals could either count extremely fast without concern for accuracy or extremely accurate without concern for speed. The general incentive to count both fast and accurate equally for the two counting methods should have induced participants to value both of these qualities as optimally as possible under experimental conditions. Finally, the benefits of the SAFEcount device appear to be most pronounced in the counting of large prescriptions, e.g. 60 and 120 pills. As there is little data on this topic in the literature, we do not know what proportion of prescriptions currently filled are of this larger variety. This should be an area for future research.

## Conclusions

Despite participants relatively low experience with SAFEcount, this head-to-head comparison of two methods for counting pills for prescriptions showed that SAFEcount pill counting was 2.6 times more accurate and 42% faster than traditional hand counting. SAFEcount could be a valuable tool for improving the quality of pharmaceutical dispensing in a wide array of settings which currently rely on traditional hand counting of bulk prescriptions.

## Supporting information

**S1 Photograph. Photograph 1: Sideview of SAFEcount device.**
(TIF)

**S2 Photograph. Photograph 2: Top view of SAFEcount device.**
(TIF)

**S3 Photograph. Photograph 3: Handle view of SAFEcount device.**
(TIF)

## Acknowledgments

We wish to thank the clinic supervisors and study participants for making this study possible. We also thank the many donors to Opportunity Solutions International for their support.

## Author Contributions

**Conceptualization:** Paul J. Krezanoski, Joseph D. Krezanoski, Alison B. Comfort, Kidwell Matshotyana.

**Formal analysis:** Paul J. Krezanoski, Joseph D. Krezanoski, Rose Gabert, Alison B. Comfort.

**Funding acquisition:** Paul J. Krezanoski, Alison B. Comfort.

**Investigation:** Joseph D. Krezanoski, Phinda Khumalo.

**Methodology:** Paul J. Krezanoski, Nkosinathi Nxumalo, Alison B. Comfort, Phinda Khumalo, Kidwell Matshotyana.

**Project administration:** Paul J. Krezanoski, Joseph D. Krezanoski, Phinda Khumalo, Kidwell Matshotyana.

**Supervision:** Paul J. Krezanoski, Nkosinathi Nxumalo, Kidwell Matshotyana.

**Writing – original draft:** Paul J. Krezanoski, Joseph D. Krezanoski.

**Writing – review & editing:** Paul J. Krezanoski, Joseph D. Krezanoski, Nkosinathi Nxumalo, Rose Gabert, Alison B. Comfort, Phinda Khumalo, Kidwell Matshotyana.

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
