## [Decision Letter · Decision Letter 0]

10 Jul 2019

PONE-D-19-15233

Comparison of traditional methods versus SAFEcount for filling prescriptions: a pilot study of an innovative pill counting solution in eSwatini

PLOS ONE

Dear Dr. Krezanoski,

Thank you for submitting your manuscript to PLOS ONE. After careful consideration, we feel that it has merit but does not fully meet PLOS ONE’s publication criteria as it currently stands. Therefore, we invite you to submit a revised version of the manuscript that addresses the points raised during the review process.

We would appreciate receiving your revised manuscript by Aug 24 2019 11:59PM. To enhance the reproducibility of your results, we recommend that if applicable you deposit your laboratory protocols in protocols.io, where a protocol can be assigned its own identifier (DOI) such that it can be cited independently in the future. For instructions see: http://journals.plos.org/plosone/s/submission-guidelines#loc-laboratory-protocols

We look forward to receiving your revised manuscript.

Kind regards,

Simone Borsci, Ph.D.

Academic Editor

PLOS ONE

**Journal Requirements:**

2. We note that  Figure(s) 1 in your submission contain [map/satellite] images which may be copyrighted. All PLOS content is published under the Creative Commons Attribution License (CC BY 4.0), which means that the manuscript, images, and Supporting Information files will be freely available online, and any third party is permitted to access, download, copy, distribute, and use these materials in any way, even commercially, with proper attribution. For these reasons, we cannot publish previously copyrighted maps or satellite images created using proprietary data, such as Google software (Google Maps, Street View, and Earth). For more information, see our copyright guidelines: http://journals.plos.org/plosone/s/licenses-and-copyright.

a) You may seek permission from the original copyright holder of Figure(s) [#] to publish the content specifically under the CC BY 4.0 license.  

3. Please report the patent registration number in the main text (at the moment it is in the "authors comments" box)

"PJK and ABC are Founding Directors and members of the Board of Opportunity Solutions International (unpaid). JDK is an unpaid employee of Opportunity Solutions International and the inventor of SAFEcount."

**Comments to the Author**

1. Is the manuscript technically sound, and do the data support the conclusions?

Reviewer #1: Yes

Reviewer #2: Partly

2. Has the statistical analysis been performed appropriately and rigorously? 

Reviewer #1: Yes

Reviewer #2: Yes

3. Have the authors made all data underlying the findings in their manuscript fully available?

Reviewer #1: No

Reviewer #2: No

4. Is the manuscript presented in an intelligible fashion and written in standard English?

Reviewer #1: Yes

Reviewer #2: Yes

5. Review Comments to the Author

Reviewer #1: A very well designed pilot study analysing accuracy and speed of a new low-cost pill counting device in low-resource settings, compared against the traditional counting method.

Manuscript is well written. I have a few comments for consideration in terms of Methods/ Discussion:

1. Did the participants have any previous exposure to SAFEcount technology?

2. How was error determined in each counting session? Was assessment blinded to the type of clouting method used?

3. How was the time spent in each counting method determined, given the protocol describes a continuous counting practice?

4. Discuss training required (i.e. the practice session) and potential impact on the scaling up of the tool

5. The new tool seems most beneficial for larger prescriptions - the two methods had little difference in either accuracy or speed for 30-pill prescription - what is the overall proportion of larger prescriptions?

Overall, a well written and interesting study showing clear benefit using the new counting approach

Reviewer #2: The authors claim that the problem of counting pills is prevalent in developing countries but there I no evidence presented by the authors on existing work in this field. Has there been any work on other pill counting devices ? It would be useful for the authors to summarise existing work in the area and draw useful insights, lessons learned etc.

There is no information on how the device has been designed. Who are the stakeholders for this device and have they been included in the design process? Was this device specifically designed for the context of use of CTX prescriptions in eSwatini or can it be used in other healthcare contexts? Are there any off the shelf device with similar functionality that could have been used instead of SafeCOUNT? What differentiates SafeCOUNT from other pill counting devices?

The photographs of the device were not available to me at the time of review so I haven’t had an opportunity to look at the device.

In regards to the design of the study, the authors compared the use of SafeCOUNT against the use of ‘traditional pill counting’. Has there been a comparison of the use of SafeCOUNT against other pill counting devices ?

The sample size for the study is quite small. 31 participants from 13 difference healthcare facilities is not sufficient to draw conclusions and generalise. 10 of the participants taken part in the study are cleaners. Is there a reason why a cleaner would be counting pills?

When the authors mention that the use of this device will result in ‘significant personnel cost savings’, how did they come up with this conclusion? A health economic study should be carried out to evaluate whether this device would be cost effective in this healthcare setting.

Overall, I think the authors should strengthen their data by conducting more studies with more people, and compare the use of their device against other existing devices. A preliminary cost analysis of introducing this device to this healthcare setting would also strengthen their argument

6. PLOS authors have the option to publish the peer review history of their article (what does this mean?). If published, this will include your full peer review and any attached files.

Reviewer #1: No

Reviewer #2: No

---

## [Author Response · Author response to Decision Letter 0]

9 Sep 2019

As included in the uploaded file:

Journal Requirements:

1) We have endeavored to ensure that the style matches the PLOS ONE format as listed. 

2) Thank you for bringing our attention to this copyright issue. We were unaware of it. We have decided to remove this map/graphic (Figure 1), as we don’t feel that it adds much more information to the overall paper. Figures and numbering throughout the manuscript have been updated accordingly.

3) As requested, we have included the patent registration number in the main text when we introduce the SAFEcount invention.

4) We agree with and are happy to add the following statement to our competing interest statement: “This does not alter our adherence to PLOS ONE policies on sharing data and materials.”

Comments to the Author

3) We have made the data utilized in this study available at the ORF public data repository. We have revised the data availability statement in the Manuscript to read: “The datasets used and/or analyzed during the current study are available at the Open Science Framework (OSF) public data repository via ‘https://osf.io/38nyj/’.”

Reviewer #1

1. This is an important question for interpreting the study results. We have included a new sentence in the methods section where we discuss the study protocol on page 3 of the track changes version: “None of the study participants had any prior experience with the SAFEcount device.”

2. Thank you for pointing out this important missing point. We have updated the methods section (again on page 3 of the track changes version) with a description of how we tracked counting errors: “Counting accuracy was determined by a second research assistant using a specially designed SAFEcount grid that was walled off to capture all pills from the pill bag. A second verification count was performed whenever an error was detected.”

3. As above, we have updated the methods section to describe how we measured the prescription times, as follows: “During the counting procedure, one research assistant recorded the prescription speeds, from pouring pills to sealing the prescription bag.”

4. We appreciate this point about the implications of training requirements on device scaling. We have added in the limitations section, where we discuss the study protocol, a discussion of some considerations related to necessary training as follows: “In scaling the device, we would expect that a longer training period, even one hour, for example, which could be self-directed, would go a long way towards decreasing errors and increasing speed even further.”

5. Indeed, we agree that the SAFEcount device appears to be most beneficial in counting larger prescriptions. Unfortunately, we are unaware of data on the relative proportion of larger versus smaller quantity prescriptions in Swaziland or elsewhere. As a result, we have added the following in the limitations section: “Finally, the benefits of the SAFEcount device appear to be most pronounced in the counting of larger prescriptions, e.g. 60 and 120 pills. As there is little data on this topic in the literature, we do not know what proportion of prescriptions currently filled are of this larger variety. This should be an area for future research.”

Reviewer #2

1. “What data do you have to support your claim of the prevalence of pill counting problems in developing countries?” This is an important point, however there are no similar studies, to our knowledge, of the problem of pill counting in developing countries. We have done extensive literature searches and have been unable to find comparable data. We have begun performing our own research, including a qualitative study that was performed in conjunction with this head to head comparison, to answer this important point. There has not been comparable work on other pill counting solutions. We discuss in the second paragraph the reasons that electronic pill counters and other technological solutions that rely on electricity and expensive parts are unsuitable to the “mid-level” environments in which we hope that the SAFEcount device can add value. 

2. “Who are the stakeholders for this device and were they included in the design?” Thank you for bringing up the importance of including local stakeholders in the design of health technologies. We believe that the design of health technologies for low resource setting should proceed with as much local knowledge integrated across the solution timeline as possible. The design of this device was achieved with extensive stakeholder input by Joseph Krezanoski when he was a Peace Corps Volunteer in Swaziland. We are planning on producing a future manuscript which discusses the community-based design of the current device and how it fits into affordable health technology best practices in general. The discussion of the design of the device was outside the scope of this head to head comparison of the device with the predominant current counting method in Swaziland clinics, and for that reason we did not discuss it and don’t think it fits into this paper at this time. We are willing to include this discussion it if it is felt to be of significant value, but we felt that it distracted from the focused study presented.. 

3. “Was SAFEcount designed only for CTX?” Thank you for pointing out that this was not clear. We discuss in the introduction of the device that SAFEcount is adaptable to other pill types, sizes, dosages and medications when we have written: “The grids can be made to function for all tablet shapes, sizes and dosages.” SAFEcount was not designed specifically for CTX, instead we used this as the prototypical pill used for bulk counting. It will function in all healthcare settings and with all medications that come in tablet or pill form. 

4. “Are there other off the shelf pill counting devices available?” To our knowledge, there are no other off-the-shelf devices that we have been able to find available in Swaziland to improve on pill counting. The devices that we have been able to find in developed countries or online are either too expensive for poor settings or unavailable in bulk. Furthermore, none of these other pill counting solutions have been rigorously compared to the current traditional hand counting as we have done in this paper.

5. “What differentiates SAFEcount from other pill counting devices?” This is a crucial point. We discuss in the introduction that other pill counting devices require electricity, expensive parts and are otherwise unsuitable for the resource limited settings in which we hope to introduce SAFEcount. Our device is designed to be “less expensive than electronic pill counters and far more accurate and rapid than traditional pill counting”.

6. “Has there been a comparison of SAFEcount against other pill counting solutions?” We have not compared SAFEcount to other pill counting solutions, for similar reasons to those discussed above. There are no other comparable solutions that are low cost, easy to use and do not require electricity and expensive parts. The motivating factor behind the design and development of SAFEcount was to make a device that was less expensive and more accessible than the electronic and other solutions that are used in the developed world.

7. “The sample size is quite small.” In terms of the sample size, we powered our study to be able to detect a significant difference between the speed of traditional counting compared to SAFEcount counting of prescriptions. We feel that the 31 participants, each counting multiple prescriptions using the two different modalities, was adequate to prove our hypothesis that SAFEcount is significantly (p<0.0001) faster than traditional hand counting.

8. “10 of the participants are cleaners.” As we discuss in the third paragraph of the Introduction, the large volumes of patients often require clinics to utilize their staff in multiple ways. Staff hired primarily to clean the facility are often (33% of our 31 participants) used also to help count patient prescriptions to meet patient demand. We included all clinic staff that counted pills as any part of their duties, as we discussed in the Methods section: “Individual pill counters were enrolled if they counted pills for patient prescriptions at least twice in the last month.”

9. “When the authors claim that the device will lead to ‘significant personnel cost savings’, how did they come up with this conclusion? A health economic study should be carried out…” We tried to be clear that we concluded that a 42% faster counting rate could lead to personnel cost savings. We explained that we believed that such an advantage could save the average pill counter 10 hours per week and that this could surely be used by the facility either to cut costs or apply those personnel hours towards other tasks. We agree that a health economic study would be of significant value, as we have recently launched just such a study.

10. We agree that getting more people to use SAFEcount, doing a cost effectiveness study and comparing it to other pill counting solutions are excellent next steps for our work.

---

## [Decision Letter · Decision Letter 1]

11 Oct 2019

Comparison of traditional methods versus SAFEcount for filling prescriptions: a pilot study of an innovative pill counting solution in eSwatini

PONE-D-19-15233R1

Dear Dr. Krezanoski,

We are pleased to inform you that your manuscript has been judged scientifically suitable for publication and will be formally accepted for publication once it complies with all outstanding technical requirements.

With kind regards,

Simone Borsci, Ph.D.

Academic Editor

PLOS ONE

Additional Editor Comments (optional):

Reviewers' comments:

Reviewer's Responses to Questions

**Comments to the Author**

1. If the authors have adequately addressed your comments raised in a previous round of review and you feel that this manuscript is now acceptable for publication, you may indicate that here to bypass the “Comments to the Author” section, enter your conflict of interest statement in the “Confidential to Editor” section, and submit your "Accept" recommendation.

Reviewer #1: All comments have been addressed

Reviewer #2: All comments have been addressed

2. Is the manuscript technically sound, and do the data support the conclusions?

Reviewer #1: Yes

Reviewer #2: Yes

3. Has the statistical analysis been performed appropriately and rigorously? 

Reviewer #1: Yes

Reviewer #2: Yes

4. Have the authors made all data underlying the findings in their manuscript fully available?

Reviewer #1: Yes

Reviewer #2: Yes

5. Is the manuscript presented in an intelligible fashion and written in standard English?

Reviewer #1: Yes

Reviewer #2: Yes

6. Review Comments to the Author

Reviewer #1: I'm satisfied that the authors have addressed my comments to the earlier version of the paper. Therefore I am happy to accept the version as it is.

Reviewer #2: The author(s) have addressed all the reviewers' comments apart from the justification of the small size of participants. Power and sample size calculations need to be included in the paper.

Once this edit has been done, I am happy to recommend the publication of this article

7. PLOS authors have the option to publish the peer review history of their article (what does this mean?). If published, this will include your full peer review and any attached files.

Reviewer #1: No

Reviewer #2: No

---

## [Editor Report · Acceptance letter]

12 Nov 2019

PONE-D-19-15233R1 

Comparison of traditional methods versus SAFEcount for filling prescriptions: a pilot study of an innovative pill counting solution in eSwatini 

Dear Dr. Krezanoski:

I am pleased to inform you that your manuscript has been deemed suitable for publication in PLOS ONE. Congratulations! Your manuscript is now with our production department. 

With kind regards,

on behalf of

Dr. Simone Borsci 

Academic Editor

PLOS ONE